# Signatures of Positive Selection in the Genome of *Apis mellifera carnica*: A Subspecies of European Honeybees

**DOI:** 10.3390/life12101642

**Published:** 2022-10-19

**Authors:** Qiang Huang, Yong-Qiang Zhu, Bertrand Fouks, Xu-Jiang He, Qing-Sheng Niu, Hua-Jun Zheng, Zhi-Jiang Zeng

**Affiliations:** 1Honeybee Research Institute, Jiangxi Agricultural University, Zhimin Ave. 1101, Nanchang 330045, China; 2Jiangxi Province Key Laboratory of Honeybee Biology and Beekeeping, Jiangxi Agricultural University, Zhimin Ave. 1101, Nanchang 330045, China; 3Shanghai-MOST Key Laboratory of Health and Disease Genomics, Chinese National Human Genome Center at Shanghai and Shanghai Institute for Biomedical and Pharmaceutical Technologies, Shanghai 200237, China; 4Institute for Evolution and Biodiversity, Molecular Evolution and Bioinformatics, Westfälische Wilhelms Universität, 48149 Münster, Germany; 5Apiculture Science Institute of Jinlin Province, Yuanlin Rd., Jinlin 132108, China

**Keywords:** honeybee, subspecies, selection, sociality, cold tolerance

## Abstract

The technology of long reads substantially improved the contingency of the genome assembly, particularly resolving contiguity of the repetitive regions. By integrating the interactive fragment using Hi-C, and the HiFi technique, a solid genome of the honeybee *Apis mellifera carnica* was assembled at the chromosomal level. A distinctive pattern of genes involved in social evolution was found by comparing it with social and solitary bees. A positive selection was identified in genes involved with cold tolerance, which likely underlies the adaptation of this European honeybee subspecies in the north hemisphere. The availability of this new high-quality genome will foster further studies and advances on genome variation during subspeciation, honeybee breeding and comparative genomics.

## 1. Introduction

Insect pollination is essential for maintaining the balance of the ecosystem, which contributes approximately 35% to crop pollination [1]. The honeybee *Apis mellifera* is a key managed pollinator, with an estimated annual global economic value of $195 billion [2]. The recent decline of honeybee colonies has provoked serious concerns regarding the biodiversity, as well as food security [3]. A number of stressors have been proven to cause the honeybee collapses, including parasites, pesticide, climate change and habitat loss [4,5,6,7,8]. Among those stressors, parasites are a major cause leading to colony losses, with a synergistic effect with pesticide. It is known that honeybee strains showed variation in tolerance towards parasites and climates [9,10,11,12,13]. However, the genetic mechanism underlying that tolerance is yet not fully understood. In this study, we aim to reveal the genomic variation within *A. mellifera* species, which is essential to understand their evolution, adaptation to different climates, as well as to refine breeding strategies for disease-tolerant strains.

In this study, we provided a highly contiguous genome assembly of a valuable subspecies of the European honeybee, *Apis mellifera carnica*. Two complementary sequencing technologies of HIFI and Hi-C were used to generate a high-quality chromosome genome assembly. Comparative genomic analysis revealed gene families selected during the social evolution and climate adaptation.

## 2. Material and Methods

### 2.1. DNA Extraction and Sequencing Library Preparation

The honeybee *A. mellifera carnica* were collected from the national honeybee breeding center in Apicultural Science Institute of Jilin, China. Six *A. mellifera carnica* drone pupae were collected from a single colony, which were pooled for DNA extraction by AxyPrep^TM^ Multisource Genomic DNA Miniprep Kit (Axygen, Irvine, CA, USA). The Qubitfluorimetry system was used to define the DNA concentration (Thermo Fisher, Waltham, MA, USA). Fragment size distribution was assessed using the Agilent 2100 Bioanalyzer with the 12,000 DNA kit (Agilent, Santa Clara, CA, USA). Then, 5 µg of high molecular weight genomic DNA was used to prepare the library. The DNA which uses g-Tube (Covaris, Woburn, MA, USA) to shear into 10 kb was used as input into the SMRTcell library preparation according to PacBio 10 kb library preparation protocol. The library was sequenced on a PacBio Sequel system using Sequencing Kit 3.0.

Additionally, five drone pupae from the same colony were fixed with formaldehyde and lysed for Hi-C library. The cross-linked genomic DNA was digested with Hind III overnight. Sticky ends of the genomic DNA were biotinylated and proximity-ligated to form chimeric junctions that were enriched for and then physically sheared to a size of 300–700 bp. Chimeric fragments representing the original cross-linked long-distance physical interactions were then processed into paired-end sequencing libraries and sequenced on the Illumina HiSeq X Ten platform.

### 2.2. Genome Assembly and Gene Annotation

The reads were filtered through Fastp with default parameters, and 2,602,139 clean Pacbio reads with a total size of 22,818,161,032 bp were obtained [14]. The reads were assembled by HGAP4 of SMRT Link (version 6.0) with default parameters. Additionally, 280,437,292 Hi-C reads with a total size of 84.13 Gb was obtained. The raw contigs were split into segments of 50 Kb on average. The Hi-C reads were aligned to the segments using BWA (version 0.7.10) with default parameters [15]. The uniquely mapped reads were retained to assemble the genome using LACHESIS package [16].

Augustus (v3.3.2) was used to train and predict the gene features with Amel_HAv3.1 gene set [17,18]. Based on the deduced amino acid sequences, the annotation was performed through BLASTP against the non-redundant peptide database with the cut-off E-value at 10^−5^ and RPS-BLAST against the Conserved Domain Database at E-value at 10^−3^ [19]. Gene ontology analysis was performed using BLASTP against the InterProScan [20]. The Pathway was constructed based on the KEGG database [19].

### 2.3. The Genome Completeness and Honeybee Evolution

The protein sequences were used to query the BUSCO arthropod ortholog set to evaluate the genome completeness. The genomes of honeybee subspecies *A. mellifera mellifera* and *A. mellifera* DH4 were further queried to *A. mellifera carnica* using minimap2 [21]. The alignment files were viewed using package pafr (https://github.com/dwinter/pafr, accessed on 7 January 2021). The microsatellite was identified using MISA (microsatellite identification tool) with default parameters [22]. The microsatellite markers primers were blasted against the three genomes and the congruent was analyzed using Chi-squared test, R [23].

### 2.4. Phylogenetic Analysis of Social Genes

To investigate the molecular mechanisms underlying sociality between social and solitary bees, the protein sequences of four insulin family proteins *insulin receptor substrate 1* (IRS1), *insulin receptor substrate 4* (IRS4), *insulin receptor like* (IR-like) and *insulin-like peptides* (IPR) were retrieved from Eastern honeybees (*Apis cerana*), dwarf honeybees (*Apis florea*), giant honeybees (*Apis dorsata*), bumble bees (*Bombus terrestris*), digger bees (*Habropoda laboriosa*), red mason bees (*Osmia bicornis bicornis*) and small carpenter bees (*Ceratina calcarata*) by NCBI blast with E-value cutoff ≤ 1 × 10^−5^. The protein sequences were aligned using Muscle with default parameters by MEGA X package (Version 10.0.2) [24]. The phylogenetic tree was constructed using the Neighbor joining model with 1000 bootstraps and the small carpenter bees were used to root the tree.

### 2.5. Identifying Genome Selection Pressures

Orthologs between 7 species (*Nasonia vitripennis*, *Bombus terrestris*, *Bombus impatiens*, *Apis florea*, *Apis dorsata*, *Apis laboriosa*, *Apis cerana*) and 4 subspecies (*Apis mellifera carnica*, *Apis mellifera caucasica*, *Apis mellifera ligustica*, *Apis mellifera mellifera*) were discovered using Orthofinder (v2.5.2) [25]. To optimize the number of single-copy orthologs, we categorized them as such if at least 3 *Apis mellifera* subspecies and *Apis cerana* had a single-copy gene for the given orthogroup, culminating at 6328 single-copy ortholog families. Phylogenetic tree reconstruction, including all species described above, was undertaken by OrthoFinder. For each single-copy ortholog family, the longest protein isoforms for each of the species’ gene were used in multiple sequence alignment with MAFFT (using local-pair algorithm and 1000 iterations) [26] and unreliably aligned residues and sequences were masked with GUIDANCE (v2.02) [27]. To optimize alignment length without gaps, we ran a maxalign script and removed subsequent sequences leading to more than 30% of gapped alignment as long as it did not result in the removal of any *A. mellifera* subspecies and *A. cerana* [28]. The protein sequences were replaced with coding sequences in the multiple alignments using the pal2nal script [29]. Furthermore, sequences containing a stop codon or having a length inconsistency between protein and DNA coding sequences (after removal of undefined bases) were filtered out. Alignments regions, where gapped positions were present, were removed with a custom python script, as these are the most problematic for positive selection inference [30,31]. Finally, CDS shorter than 100 nucleotides were eliminated [32].

Phylogenetic tests of positive selection in protein-coding genes usually contrast substitution rates at non-synonymous sites to substitution rates at synonymous sites taken as a proxy to neutral rates of evolution. The adaptive branch-site random effects model (aBSREL) from Hyphy software package was used to detect positive selection experienced by a gene family in a subset of sites in a specific branch of its phylogenetic tree [33]. The test for positive selection was run only on the branches leading to the origin of *A. mellifera* and on each *A. mellifera* subspecies. Results from the adaptive branch-site random effects model were corrected for multiple testing as one series using False Discovery Rate (FDR) and set up our significant threshold at 10% [34].

### 2.6. Test for Functional Category Enrichment

Gene Ontology (GO) annotations for our gene families were taken from Hymenoptera Genome database [35]. The enrichment of functional categories was evaluated with the package topGO version 2.4 of Bioconductor [36,37]. To identify functional categories enriched for genes under positive selection, strengthened, and relaxed selection pressure, the SUMSTAT test was used [38,39]. The SUMSTAT test is more sensitive than other methods and minimizes the rate of false positives [40,41,42,43]. To be able to use the distribution of log-likelihood ratios of the aBSREL and RELAX tests as scores in the SUMSTAT test, a fourth root transformation was used [39]. This transformation conserves the ranks of gene families [44]. Gene Ontology categories mapped to less than 10 genes were discarded. The list of significant gene sets resulting from enrichment tests is usually highly redundant. We therefore implemented the “elim” algorithm from the Bioconductor package topGO, to decorrelate the graph structure of the Gene Ontology. To account for multiple testing, the final list of *p*-values resulting from this test was corrected with the FDR and set up our significant threshold at 20%. To cluster the long list of significant functional categories, we used REVIGO with the SimRel semantic similarity algorithm and medium size (0.7) result list [45,46].

## 3. Results and Discussion

### 3.1. The Genome Assembly Statistics of A. mellifera carnica

A robust genome of 226.02 Mbp comprised of 313 contigs was assembled, which were further collapsed into 169 scaffolds (GCA_013841245.2) (Table 1). By aligning the predicted protein sequences to 1066 core arthropod Benchmarking Universal Single-copy orthologs (BUSCOs) [47], 93.53% of complete BUSCOs were identified. The results suggest that the assembled genome and predicted gene set were complete. Phylogenetic tree reconstruction revealed the subspeciation events of the European honeybees and the topology agreed with the evolution from solitary to social living (Figure 1) [48,49].

### 3.2. Genome Alignment among Honeybee Subspecies

By pair-wised alignment, 83% and 89% of *A. mellifera carnica* genome can be perfectly aligned to *A. mellifera mellifera* and *A. mellifera* DH4 genome, respectively (Figure 2 and Appendix A). The average length of the aligned region was 103 Kbp and 108 Kbp for *A. mellifera mellifera* and *A. mellifera* DH4, respectively (Appendix A). The inverted fragment may reflect natural structural variation among the genomes, reflecting local adaption [50,51]. Overall, 88,380 microsatellites with dinucleotides motif were identified in *A. mellifera carnica*. Comparatively, 89,099 and 89,569 were identified in *A. mellifera* DH4 and *A. mellifera mellifera,* respectively. The relative abundance of microsatellites along the motifs were not significantly different among the three genomes (Pearson’s Chi-squared test, df = 24, *p* = 0.26). However, the number of microsatellites decreased with the increasing number of repeats for all three genomes (Pearson’s correlation coefficient, df = 9, *p* < 0.001, Figure 3A). A set of linkage map makers were further compared among the three genomes [52,53,54]. Out of 1081 paired microsatellite primers, 839 (77%) could be aligned to all three genomes (Appendix A), with an average density of 4.8 cM per locus (Figure 3B). For the remaining 242 markers, 162 were aligned to at least one genome. *A. mellifera* DH4 shared a higher number of markers with *A. mellifera carnica* compared with *A. mellifera mellifera*, which significantly deviated from random (Pearson’s Chi-squared test, df = 2, *p* < 0.01) and is congruent with the genome phylogenetic tree in general (Appendix A).

### 3.3. Phylogenetic Analysis of Sociality-Related Proteins

The evolutionary process of bee sociality is fascinating, and highlights how genomes evolved to give rise to new and complex behaviors [49,55,56]. Insulin is an essential gene family regulating honeybee caste determination [57,58]. Hexamerin regulates the reproductive tissue development after honeybee caste differentiation [59,60,61]. The two gene families were selected to indicate the social evolution. The phylogenetic tree of insulin and hexamerin gene families clearly showed that solitary bees (digger bees and red mason bees) were an early branch from the root (small carpenter bees), followed by bumble bees (Figure 4). The four honeybee species were clustered together, indicating that the a distinctive gene selection of sociality [62,63,64].

### 3.4. Signature of Positive Selection

Overall, 4897 single-copy orthologous groups were identified, out of which 245 orthologous groups displayed signs of positive selection in at least one branch test. The number of orthologous groups under positive selection within each branch varied significantly (Chi-squared test, *p* < 2.2 × 10^−16^) and ranged from 10 to 114 with 10, 27, 45, 78, and 114 for *A. mellifera caucasica, A. mellifera mellifera*, *A. mellifera* spp., *A. mellifera carnica,* and *A. mellifera ligustica*, respectively (Appendix A). Such a variation in the number of genes under positive selection among the different *A. mellifera* subspecies may highlight a fast pace of adaptation or directed domestication in *A. mellifera carnica* and *A. mellifera ligustica* compared with others [65,66]. However, this result could also be the consequence of different population structures among *A. mellifera* subspecies with differential gene flow and introgression levels. Only one orthologous group, encoding for the protein obscurin involved in myogenesis and Hippo signaling pathway [67,68], was found to be under positive selection in three branches (*A. m. carnica*, *A. m. ligustica*, and *A. m.* spp., Table 2). Moreover, a few orthologous groups were found under positive selection in more than one branch (Table 2). Additionally, Hippo signaling pathway was involved in cold temperature adaptation in bees [13]. 

### 3.5. Functional Categories Enriched of Positively Selected Genes

We identified 11 significant functional categories in *A. m. carnica*, 28 in *A. m. ligustica*, 5 in *A. m. mellifera* and *caucasica*, and 6 in *A. mellifera* branch, which were enriched in genes under positive selection at 20% FDR. The long list of significant GO-terms found to be significantly enriched of positively selected genes in *A. m. ligustica* were mainly related to larval development (Figure 5), as demonstrated with clustering from REVIGO [45]. Interestingly, the two most significant enriched functional genes under positive selection in *A. mellifera* branch, chitin metabolism and mitochondrial translation (Figure 6), matched functional genes previously found in *A. mellifera* [34]. While functions found to be enriched of positively selected genes in *A. m. mellifera* were mainly related to the nervous system, in *A. m. caucasica.* They were mainly related to autophagy/cell death (Figure 6). In *A. m. carnica*, most of significant GO terms seems to be linked with stress tolerance, notably response to wounding, reactive oxygen species (ROS) metabolism, and larval midgut programmed cell death (Figure 6). More precisely, several significant GO terms are likely involved in cold resistance in honeybees, such as developmental growth [69], ROS metabolism [70], and hippo signaling [13]. Interestingly, the gene encoding the protein Dachsous, which was under positive selection in *A. m. carnica* and found in several significant GO terms, plays a key role in the adaptation to temperate climate in the *A. m. sinisxinyuan* [13]. Furthermore, the genes *Amel_mTOR* and *Amel*_*Imp*, encoding serine/threonine-protein kinase mTOR and insulin-like growth factor 2 mRNA-binding protein 1, respectively, might also play a role in cold tolerance as they both are involved in the insulin pathway regulating food intake, essential for cold tolerance [71,72]. Moreover, cold tolerance in *A. cerana* involved serine/threonine-protein kinases like *mTOR* [73]. The protein RB1-inducible coiled-coil protein 1, part of the GO term ‘larval midgut cell programmed cell death’, was found to be under positive selection in *A. m. caucasica* and involved in cold adaptation in amphipods [74].

## Figures and Tables

**Figure 1 life-12-01642-f001:**
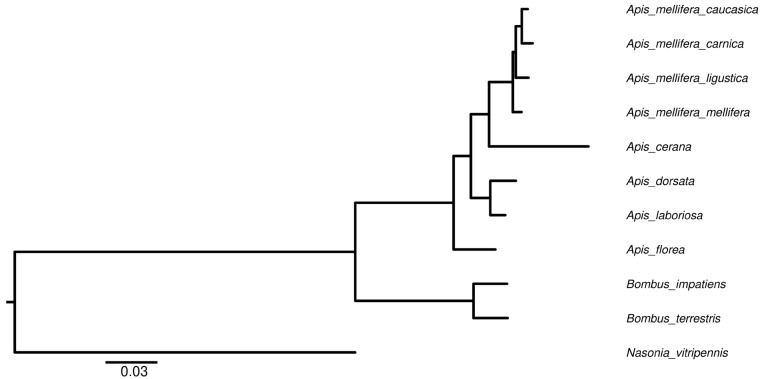
Phylogenetic tree of the studies insects for the genome selection. All nodes were 100% bootstrap supported. *N. vitripennis* was used to root the tree. For *A. mellifera carnica*, 93.53% of complete BUSCOs were found, which suggests the assembly is complete.

**Figure 2 life-12-01642-f002:**
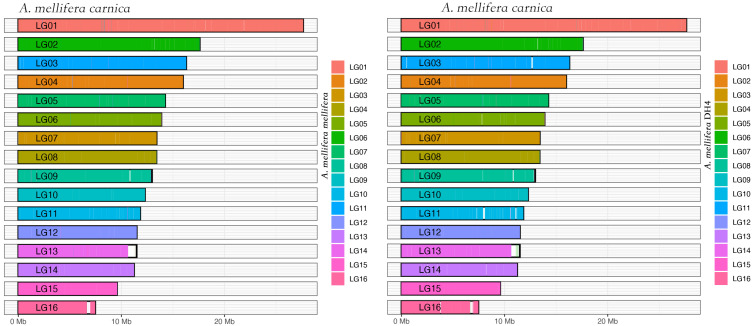
Coverage map of the honeybee subspecies. The genomes of *A. mellifera melllifera* and *A. mellifera* DH4 were compared with *A. mellifera carnica*. Overall, 80% of the nucleotides were aligned.

**Figure 3 life-12-01642-f003:**
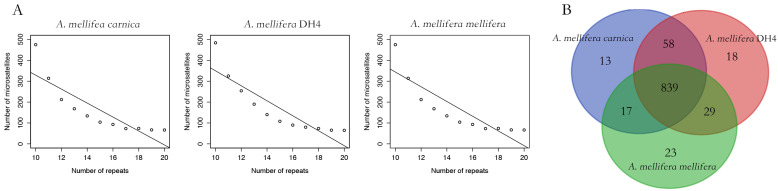
Microsatellite analysis of the three honeybee subspecies. (**A**): the distribution of microsatellites of the dinucleotide motif. The number of microsatellites decreased with the increasing number of repeats for all three genomes. Overall, the variation of the microsatellite distribution among the three genomes was not significant. (**B**): Venn diagram of the linkage map makers among the three honeybee genomes. *A. mellifera* DH4 shared significantly higher number of markers with *A. mellifera carnica* compared with *A. mellifera mellifera*.

**Figure 4 life-12-01642-f004:**
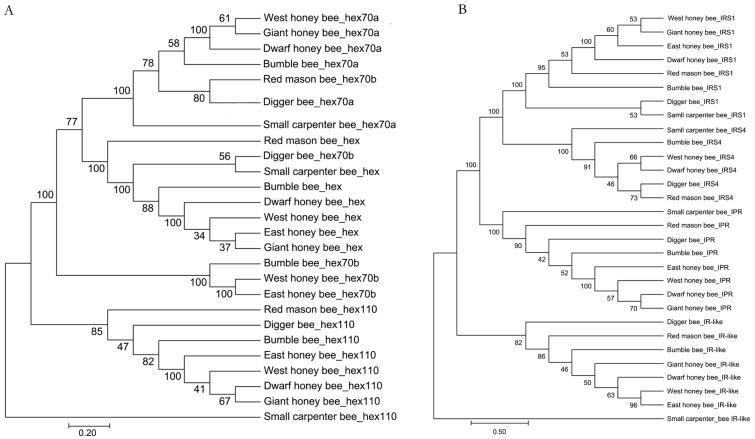
Phylogenetic tree of social genes. (**A**): phylogenetic tree of hexamerin family proteins. (**B**): phylogenetic tree of insulin family. The orthologs were retrieved by BLAST in NCBI manually with E-value cutoff ≤ 1 × 10^−5^. The sequences were aligned with Muscle and the phylogenetic tree was constructed using Neighbor joining model with 1000 bootstrap.

**Figure 5 life-12-01642-f005:**
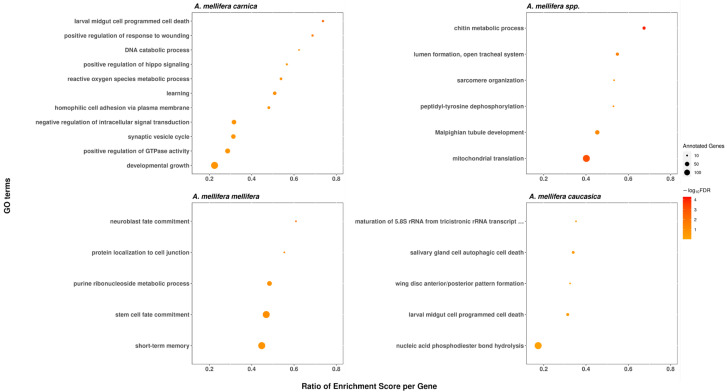
Functional categories significantly enriched of genes under positive selection in the branch *A**. m. ligustica*, using REVIGO to cluster the long list of significant GO terms. The medium cluster size was 0.7 and the semantic similarity measure SimRel.

**Figure 6 life-12-01642-f006:**
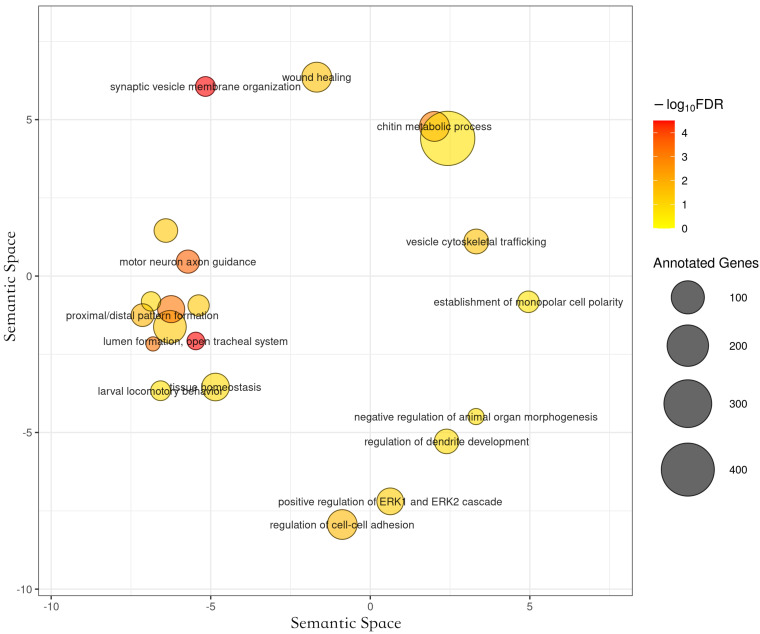
Functional categories significantly enriched of genes under positive selection in the branch *A. mellifera* spp.

**Table 1 life-12-01642-t001:** Statistics of the studied *A. mellifera carnica* genome and other recently assembled honeybee genomes. NA indicates that the genome was assembled at contig level.

	*A. mellifera* *carnica*	*A. mellifera* *caucasica*	*A. mellifera* DH4	*A. mellifera* *mellifera*	*A. cerana cerana*
Technology	Pacbio, Hi-C	Pacbio	Pacbio, Hi-C	Pacbio, Hi-C	Pacbio, Hi-C
Coverage	101	112	192	100	134
Assembly size (Mbp)	226.02	224.7	225.25	227.03	215.6
Number of Contigs	313	224	227	199	214
N50 Scaffold size (Mbp)	13.4	NA	13.6	13.5	13.7
Releasing date	2020	2020	2018	2018	2020

**Table 2 life-12-01642-t002:** Orthologous found to be under positive selection in more than one branch tested.

Gene/Protein Name	Putative Function	Branches
obscurin	Involved in myogenesis and the Hippo signaling pathway	*A. mellifera carnica* *A. mellifera ligustica* *A. mellifera* spp.
ATP-dependent RNA helicase abstrakt	axonal growth (visual system)	*A. mellifera carnica* *A. mellifera ligustica*
bifunctional heparan sulfate N-deacetylase/N-sulfotransferase	involved in Wnt signaling	*A. mellifera carnica* *A. mellifera ligustica*
foxP protein	Central Nervous System/learning-memory	*A. mellifera carnica* *A. mellifera ligustica*
serine/threonine-protein kinase tricorner	Post-transcriptional regulation, development (dendrite morphogenesis)	*A. mellifera carnica* *A. mellifera ligustica*
protein turtle homolog B	Establishing coordinated motor control, axonal targeting of the R7 photoreceptor	*A. mellifera carnica* *A. mellifera ligustica*
GTPase-activating Rap/Ran-GAP domain-like protein 3	signal transduction	*A. mellifera carnica* *A. mellifera ligustica*
zinc finger protein 512B	involved in transcriptional regulation	*A. mellifera carnica* *A. mellifera mellifera*
prohormone-2 precursor	neuropeptide/social behavior regulation	*A. mellifera carnica* *A. mellifera mellifera*
protein lingerer	copulation/short-term memory	*A. mellifera carnica* *A. mellifera mellifera*
retinoid-inducible serine carboxypeptidase	involved in vascular wall	*A. mellifera carnica* *A. mellifera mellifera*
cAMP-specific 3′,5′-cyclic phosphodiesterase	signaling, phsysiology, female fertility, learning/memory	*A. mellifera carnica* *A. mellifera mellifera*
7SK snRNA methylphosphate capping enzyme	methylation, development	*A. mellifera carnica* *A. mellifera mellifera*
RB1-inducible coiled-coil protein 1	involved in autophagy	*A. mellifera carnica* *A. mellifera caucasica*
muscle LIM protein Mlp84B	cell differentiation late in myogenesis	*A. mellifera carnica* *A. mellifera* spp.
junctophilin-1	formation of junction membrane in sarcomere	*A. mellifera carnica* *A. mellifera* spp.
RNA binding protein fox-1 homolog 2	alternative splicing	*A. mellifera ligustica* *A. mellifera mellifera*
chitin synthase chs-2	chitin biosynthetic synthesis	*A. mellifera ligustica* *A. mellifera* spp.
ell-associated factor Eaf	regulation of transcription elongation	*A. mellifera ligustica* *A. mellifera* spp.
nuclear protein localization protein 4 homolog	ubiquitination	*A. mellifera ligustica* *A. mellifera* spp.
kinesin-like protein unc-104	synaptic vesicle transport/locomotion	*A. mellifera ligustica* *A. mellifera* spp.
actin-interacting protein 1	sarcomere organization/development/locomotion	*A. mellifera ligustica* *A. mellifera* spp.
ubiquitin carboxyl-terminal hydrolase 36	stem cell line maintenance	*A. mellifera ligustica* *A. mellifera* spp.
teneurin-a	neural development	*A. mellifera ligustica* *A. mellifera* spp.
troponin I	muscle contraction	*A. mellifera ligustica* *A. mellifera* spp.

## Data Availability

The raw sequencing reads have been deposited in BioProject PRJNA644991. The genome and gene annotation file are at: https://www.ncbi.nlm.nih.gov/assembly/GCA_013841245.2 (accessed date 31 August 2022).

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
