# Peer review of "Signatures of Positive Selection in the Genome of Apis mellifera carnica: A Subspecies of European Honeybees"

_life, 2022, doi:10.3390/life12101642_

Round 1
Reviewer 1 Report
I found this study as interesting.
Line 101 - italicize the species name.
Author Response
Response: species names have italicized and we double checked through the text.
Reviewer 2 Report
Dear Dr. Authors
Please find the comments.
Abstract: Should include background, objective, materials and methods, results, and conclusion.
Introduction:
· A poor section, and needs improvement.
· The aim is not clear.
Results and Discussion
· discussion needs more improvements.
Conclusion
· There is no conclusion.
References
· Scientific names should be in italic form.
· The authors write honeybee(s) as one word and sometimes as two words. Please use one form throughout the manuscript.
Author Response
Abstract: Should include background, objective, materials and methods, results, and conclusion.
Response: Even though the journal does not restrict the abstract, we do include the background, merged objective and methods, results and outlook, a more concise format.
Introduction:
- A poor section, and needs improvement.
- The aim is not clear.
Response: we have revised the introduction section and clarified the aim as: In this study, we aim to reveal the genomic variation within A. mellifera species, which is essential to understand their evolution, adaptation to different climate, as well as to refine breeding strategies for disease tolerant strains (Line: 39-40).
Results and Discussion
- discussion needs more improvements.
Response: in this section, a few points have been improved (Line: 145, 146, 164, 178-180, 184, 186, 197-199).
Conclusion
- There is no conclusion.
Response: "conclusion" is optional. however, we do indicated:
Positive selection was identified in genes involved with cold tolerance, which is likely underlying the adaptation of this European honeybee subspecies to north hemisphere. The availability of this new high-quality genome will foster further studies and advances on genome variation during subspeciation, honeybee breeding and comparative genomics (Line 23-26).
References
- Scientific names should be in italic form.
- The authors write honeybee(s) as one word and sometimes as two words. Please use one form throughout the manuscript.
Response: thanks for point it out, which escaped our attention. we have checked through the text to italize species names. honeybee is now used through the text.
Reviewer 3 Report
Signatures of positive selection in the genome of Apis mellifera carnica: a subspecies of European honey bees is an interesting article showing a distinctive pattern of genes involved in social evolution by comparing social and solitary bees. Positive selection was identified in genes involved with cold tolerance that is probably related to the adaptation of this European honey bee subspecies to the northern hemisphere. The availability of this new high-quality genome will allow further studies and advances in genome variation during subspeciation, honey bee breeding, and comparative genomics.

Author Response
thanks
Round 2
Reviewer 2 Report
The manuscript could be accepted